# Sheep’s Butter and Correspondent Buttermilk Produced with Sweet Cream and Cream Fermented by Aromatic Starter, Kefir and Probiotic Culture

**DOI:** 10.3390/foods12020331

**Published:** 2023-01-10

**Authors:** Tânia Silva, Arona Pires, David Gomes, Jorge Viegas, Susana Pereira-Dias, Manuela E. Pintado, Marta Henriques, Carlos Dias Pereira

**Affiliations:** 1Instituto Politécnico de Coimbra, Escola Superior Agrária, 3045-601 Bencanta, Portugal; 2Centro de Estudos dos Recursos Naturais Ambiente e Sociedade (CERNAS), 3045-601 Bencanta, Portugal; 3Centro de Biotecnologia e Química Fina (CBQF)-Laboratório Associado, Escola Superior de Biotecnologia, Universidade Católica Portuguesa, Rua Diogo Botelho 1327, 4169-005 Porto, Portugal

**Keywords:** sheep, butter, aromatic, kefir, probiotics

## Abstract

Small ruminant dairy products are common in some Mediterranean countries, in the Middle East and Africa, and can play a particular role in the development of rural areas. Butter has been the object of few research studies aimed at evaluating its potential as a vehicle for probiotic microorganisms. Moreover, the recovery of fermented buttermilk with functional properties can be considered an excellent opportunity to value this dairy byproduct. Therefore, the purpose of the present work was to develop different sheep butters and respective buttermilks after cream fermentation by: (1) a mesophilic aromatic starter (A); (2) a kefir culture (K); and (3) a mixture of probiotic bacteria (P). The butters and buttermilk produced with fermented cream were compared with non-fermented sweet cream (S) butter or buttermilk, respectively, regarding their physicochemical, microbiological and sensory characteristics. The adjusted production (%, *w*/*v*) obtained for butter were: S (44.48%), A (36.82%), K (41.23%) and P (43.36%). S, A and K butters had higher solids, fat and ashes contents than P butter. The probiotic butter had a total fat of ca. 75% (*w*/*w*), below the legal limits, while all others had fat levels above 81.5%. In all samples, the pH decreased and the acidity increased over 90 days of refrigerated storage. These variations were more evident in the P butter, which agrees with the highest lactic acid bacteria counts found in this sample. Differences in color between samples and due to storage time were also observed. In general, the butter samples tended to become darker and yellower after the 60th day of storage. Texture analysis showed comparable results between samples and greater hardness was observed for the P butter, most probably due to its higher relative saturated fatty acids content (66.46% compared to 62–64% in S, A and K butters). Regarding rheological properties, all butters showed pseudoplastic behavior, but butter P had the lowest consistency index (249 kPa.s^n−1^). The probiotic butter and the corresponding buttermilk had viable cell counts greater than 7 Log CFU/g, indicating their suitability as probiotic carriers. All products were well accepted by consumers and small, but non-significant, differences (*p* > 0.05) were observed in relation to the sensory parameters evaluated. In general, it can be concluded that the use of adequate starter cultures can allow the production of innovative and potentially healthier products, alongside the valorization of dairy byproducts, improving the income of small-scale producers.

## 1. Introduction

Although the application of sheep’s or goat’s milk for cheese making is well-known, few attempts have been made to systematically study the use of such milks to produce cream, butter, yoghurt and other traditional dairy products. However, products made with sheep’s or goat’s milk, due to their nutritional and antiallergenic properties, can be a beneficial alternative in terms of health benefits [1,2]. Hayaloglu and Karagul-Yuceer [3] indicate that products made with sheep’s or goat’s milk have some different and interesting characteristics, mainly taste, aroma, appearance and chemical composition, compared with those made with cow’s milk. However, they also indicate that, in Turkey, over time, a significant decrease in the production of goats or sheep is evident, due to the migration of the population from rural to urban areas, low yields in production and milking, unproductive pastures, increased feed and labor prices, among other factors. The same evidence holds true for other countries, such as Portugal. Sustainable rural development requires value-added products and activities. A recent study by Tulla [4] discusses four opportunities to valorize peripheral rural areas: (a) transformation of dairy products into competitive value-added commodities; (b) promotion of extensive livestock based on local natural grass fodder; (c) development of value-added tourist activities linked to local landscape; and (d) planning value-added cultural activities related to cultural heritage. 

Sheep’s and goat’s butters are seldom reported in the literature, despite being common dairy products in some Mediterranean countries, in the Middle East and Africa. Information about its composition and characteristics is critically scarce. Only a few reports refer to the physicochemical and microbial characteristics of traditional butters made with cow’s cream or with sheep’s and goat’s cream [5,6,7]. As consumers’ demands for dairy products of high nutritional and health value have increased the popularity of small ruminants’ dairy products, it is important to increase the research efforts on these valuable products. In fact, the development of new functional foods is driven by the increased public awareness regarding the health impact of food [8,9]. 

Functional foods, especially probiotics, exert a beneficial effect on the gut microbiota of the host after consumption and can prevent several health problems [10]. Dairy products are traditionally used as vehicles for probiotics in the human diet. However, levels equal to, or greater than, 6–7 Log CFU/g have been suggested as the minimum concentration for positive effect of probiotic bacteria at the time of consumption. This aspect represents a challenge, as several factors during processing and storage affect the viability of these microorganisms [11,12]. A recent review comparing dairy and non-dairy fermented food products as probiotics deliver matrixes did not mention butter as a potential carrier of probiotic bacteria [13]. However, the use of probiotic bacteria in butter can represent an excellent opportunity to increase the market potential of such product.

Cream is the main raw material to produce butter and reflects its properties in butter quality. Cream fermentation with starter cultures impacts the physicochemical and sensory properties of butter and influences its nutritional composition, texture and shelf life [14,15]. Usually, cream undergoes fermentation with mesophilic lactic acid bacteria, namely *Lactococcus lactis* ssp. *dyacetilactis* and/or *Leuconostoc citrovorum* (commonly referred to as an aromatic starter) [16]. Although the use of probiotic bacteria in cream has received little attention, some authors report their use, and their results will be compared to ours. 

Buttermilk is the by-product of butter manufacture. Sweet-cream buttermilk is similar in composition to skim milk, except for its high content of phospholipids and milk fat globule membrane (MFGM) [17]. Fat globules are stabilized by the MFGM, which is composed of phospholipids, various glycoproteins, enzymes and cholesterol [18]. Several authors refer to the interesting nutritional and health properties of buttermilk [19,20,21,22,23,24,25]. 

Due to its composition and proven health benefits, several papers refer to processes and products in which buttermilk is used as the main ingredient, with the addition of several ingredients such as fruits [26] or other materials of plant origin (e.g., *Aloe vera*) [27]. Other works refer to the modification of buttermilk composition by selective concentration of its solid components [28], associated or not to protein hydrolysis [29]. Moreover, in other reports, buttermilk was used as a substrate for fermentation using conventional starters [30] or probiotic microorganisms [31,32,33,34]. The product can also be carbonated [35]. 

Buttermilk is almost unknown for European consumers. Therefore, the introduction in the market of buttermilk resulting from small ruminants’ local breeds, with adequate levels of probiotic bacteria and good sensory properties, can be an interesting option for small scale producers of rural areas. Considering the importance of the subject, in the present work we compared the physicochemical (dry matter, fat, protein, ashes, fatty acids composition, color, texture and rheological properties), microbiological (lactobacilli, lactococci, yeasts and molds) and sensory characteristics of sheep’s butters produced with sweet cream and with cream ripened with aromatic cultures, kefir and probiotics. Additionally, the characteristics of the correspondent buttermilks were also assessed and compared. 

## 2. Materials and Methods

### 2.1. Butter and Buttermilk Production

500 L of sheep’s milk, supplied by the company Rebanhos d’Avó, Soure, Portugal, were processed in the dairy pilot plant of Escola Superior Agrária, Coimbra, Portugal. The milk was produced by Lacaune ewes, aged 3 years on average, fed on pasture and supplemented with concentrate upon machine milking. The cream was obtained by means of a Westfalia Separator, type ADB (GEA Group, Oelde, Germany). The obtained cream (60 L with 40% fat) was pasteurized at 90 °C for 5 min and equally divided into four portions. Before churning, each 15 L of cream were subjected to the following conditions:(a)Non-inoculated and kept at 4 ± 2 °C for 48 h (S—sweet);(b)Inoculated with 5% (*v*/*v*) aromatic culture (Flora Danica, FD-DVS™, CHR Hansen, supplied by Promolac, Lisbon, Portugal), composed of *Lactococcus lactis* subsp. *cremoris*, *Leuconostoc*, *Lactococcus lactis* subsp. *lactis*, *Lactococcus lactis* subsp. *lactis* biovar *diacetylactis* and kept for 24 h at 20 °C, then maintained at 4 ± 2 °C for 24 h (A—aromatic);(c)Inoculated with 5% (*v*/*v*) kefir culture (eXact Kefir 1™, CHR Hansen, supplied by Promolac, Lisbon, Portugal), composed of *Debaryomyces hansenii, Lactococcus lactis* subsp. *cremoris*, *Lactococcus lactis* subsp. *lactis* biovar *diacetylactis*, *Lactococcus lactis* subsp. *lactis*, *Leuconostoc*, *Streptococcus thermophilus* and kept for 24 h at 20 °C, then maintained at 4 ± 2 °C for 24 h (K—kefir);(d)Inoculated with 5% (*v*/*v*) probiotic culture composed of a mixture of *Lactobacillus paracasei* (nu-trish L. casei 431™, CHR Hansen), *Lactobacillus acidophilus* (nu-trish LA-5™, CHR Hansen) and *Lactobacillus rhamnosus* (nu-trish LGG™, CHR Hansen, supplied by Promolac, Lisbon, Portugal), in the proportion 1:1:1 and kept for 24 h at 20 °C, then maintained at 4 ± 2 °C for 24 h (P—probiotics) [36,37,38].

All freeze-dried cultures had been previously incubated for 24 h at 35 °C in sterile milk. During cream fermentation, 15 mL samples were collected in duplicate for chemical analyses (pH and titratable acidity) right after the culture inoculation (0 h), at 12 and 24 h of fermentation to evaluate the progress of cream fermentation. Considering manufacturer’s indications regarding numbers of viable cells in freeze dried cultures, it can be estimated that the initial levels of the inoculated cultures were of the order of 5–7 Log UFC/mL cream.

Butter samples were produced with a 30 L butter churner (Termoinox, Vale de Cambra, Portugal). Butter grains were washed with cold (4 °C) pasteurized water and salted (1%, *w*/*w*), prior to the final working step, in which the butter grains were pressed and squeezed to remove the moisture between them.

Finally, the butter samples were packed into 500 mL polypropylene boxes and stored at <7 °C for 90 days (extra samples were also kept frozen at −25 ± 2 °C). Samples for analysis were collected one day after production and after 30, 60 and 90 days after production.

Butter yields were calculated as the mass of butter obtained by volume of cream (yield = mass butter/volume cream × 100). Yields were also adjusted to a reference content of total solids (85% *w*/*w*) for comparison purposes.

After cream churning, the buttermilks were collected and evaluated over 28 days of refrigerated storage.

### 2.2. Physicochemical Analyses

The butters and respective buttermilks were evaluated for dry matter, fat, protein, ash, fatty acid composition, color, texture, rheology and viscosity. All physicochemical analyses were performed at least in triplicate.

#### 2.2.1. Moisture and Total Solids

Analytical methods applied to milk were used for the analyses of buttermilk. Moisture was determined according to method 925.23 AOAC [39]. Total solids were calculated by difference.

#### 2.2.2. Ashes

Ashes were determined according to method 945.46 AOAC [40].

#### 2.2.3. Titratable Acidity

The titratable acidity of cream was evaluated according to Portuguese standard NP 638 [41], and for butter, the Portuguese standard NP-1712 [42] was applied. The titratable acidity of buttermilk was evaluated according to NP 470 [43]. 

#### 2.2.4. pH

The pH was directly determined using a pH meter (Hanna Instruments, model HI9025, Póvoa de Varzim, Portugal) with an FC200 probe. The pH meter was previously calibrated according to manufacturer’s instructions using Hanna Instruments HI7004 (pH 4.01) and HI 7007L (pH 7.01) buffer solutions.

#### 2.2.5. Fat

The fat content of butter and buttermilk samples was analyzed by the Gerber method [44].

#### 2.2.6. Protein

Protein evaluation was performed by the Kjeldahl method following the procedure described by Egan et al. [45].

#### 2.2.7. Fatty Acid Composition

Lipid fractions extracted from butter samples suspended in hexane were subjected to the following procedure for derivatization of total fatty acids: mixture of hexane/oil was transferred to borosilicate glass tubes with acid/heat resistant cap (VWR, Leuven, Belgium) at determined volume in order to achieve 15 mg of oil in the tube. The mixture was then dried under a flow of nitrogen gas until complete evaporation of the solvent. Thereafter, 100 µL internal standard tritridecanoin 99% (TG-C13; 1.7 mg mL^−1^), 900 µL of hexane, 2.26 mL of methanol and 240 µL of sodium methoxide (MetNa) were added to the oil, shaken in vortex at maximum speed for 30 s and then left at 80 °C in Multi-block Heater 2053 (Lab-Line, Melrose Park, IL, USA) for 10 min. After cooling down in ice bath, 1.25 mL of dimethylformamide (DMF) and 1.25 mL of H_2_SO_4_ at 3M diluted in methanol were added and heated at 60 °C for 30 min in multi-block heater. After cooling down in ice, 1 mL of hexane was added to the mixture, shaken in vortex for 30 s and centrifuged at 1250× *g* for 5 min. Thereafter, supernatant fraction was collected in vials for injection in GC apparatus. All the mentioned reagents were HPLC grade from VWR Scientific, Leuven, Belgium, except for MetNa (Sigma-Aldrich, St. Louis, MO, USA) and tritridecanoin (Larodan, Solna, Sweden). Fatty acid profile of butter samples was analyzed in a gas chromatograph HP 6890 (Hewlett-Packard, Avondale, PA, USA), equipped with a flame-ionization detector (GLC-FID) and a BPX70 capillary column (50 m × 0.25 mm × 0.25 mm; SGE Europe Ltd., Courtaboeuf, France). Analysis conditions were as follows: injector (split 25:1; injection volume 1 µL) and detector temperatures were 250 °C and 275 °C, respectively; flow rate was set at 1 mL/min; carrier gas was hydrogen (20.5 psi). The oven temperature program was as follows: 60 °C (held 5 min), then raised at 15 °C/min to 165 °C (held 1 min) and finally at 2 °C/min to 225 °C (held 2 min). For the individual identification of fatty acids, Supelco 37 and FAME from CRM-164 were used for individual fatty acids’ identification. In addition, GLC-Nestlé36 was assayed for calculation of response factors and detection and quantification limits. Calculation of response factors and detection and quantification limits: LOD: 0.79 μg FA/mL; LOQ: 2.64 μg FA/mL.

#### 2.2.8. Color

The color parameters of butter and buttermilk samples were determined with a Minolta Chroma Meter, model CR-200B colorimeter (Konica, Osaka, Japan) calibrated with a white standard (CR-A47: Y = 94.7; x 0.313; y 0.3204). The following conditions were used: illuminant C, 1 cm diameter aperture, 10° standard observer. Color coordinates were measured in the CIEL *a*b* system. The color difference (∆Eab*) was calculated as:∆Eab* = [(L* − L_R_*)^2^ + (a* − a_R_*)^2^ + (b* − b_R_*)^2^]^1/2^(1)
where L_R_*, a_R_* and b_R_* were the values measured for the reference sample and L*, a* and b* for the comparing samples. A matrix of ∆Eab* values between butter samples was constructed.

#### 2.2.9. Texture Profile

The texture properties of butter samples were evaluated applying a texture profile analysis test, using TA.XT, texture analyzer (Stable Micro Systems, Godalming, UK) using a conical probe 60° reference TA2. A compression test was performed with a pre-test speed of 1.0 mms^−1^, a test speed of 7.0 mms^−1^, a post test speed of 5.0 mms^−1^ and a distance of 20.0 mm. Samples with ca. 5 cm height were evaluated at a temperature of 7 ± 2 °C. The following parameters were quantified: hardness, adhesiveness, gumminess, springiness, cohesiveness and resilience.

#### 2.2.10. Rheological Analysis

Rheological tests were performed at 5 °C, using a Rheostress 6000 rheometer with a plate-to-plate (TMP35 bottom plate and P35 TiL upper plate) equipped with the software HAAKE RheoWin Job Manager software, version 4.82.0002 for data acquisition (ThermoHaake™, ThermoFisher Scientific, Waltham, MA, USA). Values for the elastic and viscous moduli (G′ and G″) and of the complex viscosity (η*) were obtained at frequencies from 0.05 to 1.0 Hz. The complex viscosity of the samples was adjusted to the Power Law or Ostwald model according to equation 2, and the consistency index K (Pa.s^n−1^) and Power-Law index (n) were determined. This model is usually applied for shear-thinning fluids such as weak gels and low-viscosity dispersions.
H* = K (γ)^n−1^(2)

#### 2.2.11. Viscosity

The viscosity of buttermilk samples (300 mL) was evaluated using a Brookfield viscometer, model DV2T (Ametek, Harlow, UK) using spindle n°1 and a speed of 40 rpm for 1 min at 7 ± 2 °C. Results are presented in centipoises (cPs).

### 2.3. Microbiological Analysis

The microbiological analysis included the quantification of lactococci, lactobacilli and yeasts and molds. From each collected sample (10 g of butter, 15 mL of buttermilk) tenfold dilutions were prepared in Ringer solution with tween 80 (Merck Schuchardt, Hohenbrunn, Germany). In the case of butter, the first dilution was prepared by melting the samples, previously placed in a sterile Stomacher’s bag with Ringer solution and tween 80, at 42 °C in a water bath. The quantification of lactobacilli was performed according to ISO 15214 [46]. Briefly, the prepared dilutions were inoculated by pour plate method in MRS agar (De Man Rogosa and Sharp-Biokar Diagnostics, Allone, France) in double layer and incubated at 37 °C for 72 h. under anaerobic conditions. Lactococci were evaluated on M17 agar (Biokar Diagnostics, Allone, France) and incubated at 37 °C for 48 h. Yeasts and molds were evaluated using Cooke Rose Bengal agar (Liofilchem Diagnostici, Roseto degli Abruzzi, Italy) according to ISO 6611 [47] and incubated at 25 °C for 7 days. All microbiological analyses were performed in triplicate. 

### 2.4. Sensory Analysis

Butter and buttermilk samples were submitted to sensory analysis by panel of 30 untrained members according to ISO 4121 [48], to simulate consumers’ perception of the products. A preference test was conducted using a 5-point hedonic scale (1—dislike a lot; 5—like a lot) for aroma, taste, texture, and global evaluation. A ranking test was also performed to confirm the results of the preference test [49]. All panel members were volunteers and gave written permission for the analysis and dissemination of results.

### 2.5. Statistical Analysis

The results were submitted to one-way ANOVA, using the STATISTICA Software V.12.0 (Statsoft, Tulsa, OK, USA). Mean values were compared using the Tukey’s honestly significant difference (HSD) test. Differences were considered significant at *p* < 0.05. 

## 3. Results and Discussion

The chemical characteristics of the butter samples are presented in Table 1. The sweet (S), aromatic (A) and kefir (K) butter samples presented more than 80% (*w*/*w*) fat, while the butter produced with cream fermented by probiotic bacteria had significantly lower values, both for total solids and fat. In this case, the value obtained for fat is clearly below the legal level (80% *w*/*w*) indicated for salted butter by the Portuguese standard, NP-1711 [50]. It was observed that the churning process of this type of butter was also significantly longer (45 min) compared to the other butters (15–30 min). The churning time varies from batch to batch, as it depends on the aggregation speed of destabilized fat globules to form butter grains. This process is empirically controlled by observing the characteristics and size of butter grains. The presence of higher levels of unsaturated fatty acids, which are liquid at the temperature, used during the churning process (10–14 °C) improves the aggregation of granules by sticking particles together. Therefore, as the P butter contained lower levels of unsaturated fatty acids (chiefly C18:1 c9), its churning process took longer and less moisture was expelled during the process, resulting in higher moisture in the final product.

The yields (% *w*/*v*) obtained for the different butter samples were as follows: S (43.33%), A (35.73%), K (40.95%) and P (46.95%). Adjusted yields were also calculated considering the legal limit of a maximum of 15% moisture: S (44.48%), A (36.82%), K (41.23%) and P (43.36%). In both cases, the lower yield obtained with cream fermented by aromatic and kefir cultures is clear. An increase in the total solids content of the buttermilks resulting from these two butters would explain the yield losses. However, the buttermilks’ compositions presented do not support this hypothesis. Most likely, the lower yields of the butters A and K can be explained by losses of fat in the walls of the cream churner. The higher levels of unsaturated fatty acids in the creams that originated these butters may have increased the adhesion of fat to the walls of the equipment, reducing the amount of fat recovered after churning. 

The percentage fat in dry matter of all samples was similar in all cases: S (95.86 ± 0.78%), A (95.25 ± 1.08%), K (95.66 ± 1.31%), and P (95.23 ± 0.67%), indicating a similar recovery of fat in all cases. Hilali and Rischkowsky [51], report that sheep’s milk butter contains 83.1–85.3% total solids and 81.4–83.2% fat, with a fat recovery rate of 94.1–97.5%, depending on processing conditions, especially temperature. These results agree with the data obtained in the present work.

The evolution of pH and titratable acidity of the butter samples over the 90 days is presented in Figure 1. In all cases, the decrease in pH and the increase in titratable acidity over time is clear. The pH and acidity of the sweet butter at the first day of storage reflect the significantly higher pH and lower acidity of sweet cream when compared to those in which starters were used. The butter produced with probiotics presented the lowest pH values. It also showed high acidity values, which are like the ones obtained in the aromatic butter, after the 60th day of storage. The lower pH and higher acidity developed in the P butter indicate the good adaptation of the strains of the lactobacilli mixture used to this matrix, which is also reflected by the high counts of such microorganisms. The lactobacilli and lactococci counts in the A butter also justify its low pH and high acidity. As expected, the S butter had the highest pH values and the lowest acidity, since the initial counts of lactobacilli and lactococci are much lower when compared to those of butters produced with the addition of starter cultures. The lactic microbiota in sweet butter result from adventitious contamination and not from the intentional addition of microorganisms. The observed acidity values are higher than those reported by Ferreira et al. for sweet and fermented butters containing probiotics [16].

The color parameters of all butters presented significant differences (*p* < 0.05) regarding lightness (L*), as well as regarding parameters a* (red-green axis) and b* (blue-yellow axis) (Table 2). Differences were observed in the same product on different days of storage and among products on the same day of storage. The lower lightness of the butter samples after the 60th day of storage can be pointed out as the greatest variation in the color parameters. Furthermore, at the end of storage, the b* values were higher, indicating an increase in the yellowness of the butter samples.

The color differences presented in Table 3 indicate more precisely the differences observed between the samples and that resulting from storage time. Considering the ∆Eab* values (Table 3A,B), it can be concluded that, in almost all cases, the storage time and the type of cream fermentation influenced the color of the butters, since in most comparisons these values are higher than 1, indicating that a common observer could detect differences between them [52]. 

When compared to the color of sweet cow’s butter and cow′s butter containing probiotics, reported by Ferreira et al. [16], the L* values are slightly higher, and b* values are significantly lower, indicating that sheep’s butter is lighter and cow’s butter more yellow. Since the yellow color of butter is dependent on the amount of β-carotene released from fat globules during churning, it can be concluded that sheep’s butter has lower amounts of this component.

Regarding the texture characteristics of the butters, non-significant differences (*p* > 0.05) were observed, except in the case of the hardness of the aromatic butter (A), which was significantly lower when compared to all others (Table 4). Despite its lower solids and fat contents, the highest average value of hardness was observed for the P butter, although without significant differences to the S and K butters. The S, A and K butters present lower amounts of saturated fatty acids (SFA) and higher amounts of monounsaturated fatty acids (MUFA) when compared to the P butter (Table 5). Unsaturated fatty acids contribute to the reduction in the hardness of fat due to their lower melting points, while the saturated fatty acids increase the hardness of fats. Thus, the variations in the fatty acid profile of butters interfere with its texture and can explain the higher value observed in the P butter. The values of hardness are significantly higher than the ones of sweet cow’s butter and cow’s butter with probiotics reported in a previous work [16].

Figure 2A shows the changes in elastic (G′) and viscous (G″) moduli as a function of frequency (Hz). It can be observed that butter samples did not follow a true gel behavior that can be observed by a power Law model (G′ = a.gb), where b (the slope in a log–log plot of G′ versus g) is equal to zero [53]. In the present study, b presented positive values, which are characteristic of weak gels and highly concentrated dispersions (0.26 for S; 0.33 for A; 0.41 for K and 0.45 for P).

Except for sweet and kefir butters, which showed a crossover point (G′ = G″) at 0.067 Hz and 0.0738 Hz, respectively, aromatic and probiotic butters showed an elastic response (G′ > G″) within the frequency range tested (0.01–1 Hz). In the case of S and K butter, a viscous response was observed (G″ > G′) for lower frequencies. The K butter sample presented the higher elastic (G′) and viscous (G″) values, which may be associated with stronger molecular interactions [53], while the aromatic butter presented the lowest values. Some authors have also reported that, when increasing the presence of mono and divalent salts, G′ and G″ also increase, probably because Na^+^ ions can create indirect cross-links with the water molecules [54], and Ca^2+^ can be responsible for a stronger binding of carboxylate-cation-carboxylate interactions that provide greater cross-linking capacity of adjacent polymer helices [55]. In our study, no significant differences were observed between the ash content of the butters. The proportion of these individual specific ions may be different between samples but was not investigated in the present work. Therefore, the rheological tests confirmed that the butter behaves more like a solid; that is, the deformations are elastic and recoverable [56].

The loss tangent (tan δ), which is the ratio of G″/G′, is another characteristic for the evaluation of the viscoelastic behavior of the butters. Values of tan δ < 1 indicate predominantly elastic behavior, while tan δ > 1 values indicate predominantly viscous behavior. In this work, as reported for polymer systems [56], it was observed that for frequencies lower than 0.1 Hz, butters behave like a dilute solution, presenting large values of tan δ (>0.3), while for higher frequencies the values of tan δ were between 0.2–0.3, revealing amorphous polymers’ behavior (data not shown).

Butters showed a shear dependent flow behavior as the complex dynamic viscosity (η*) decreased linearly with increasing frequency on a double logarithmic scale (Figure 2B). Therefore, we can conclude that butters demonstrated a non-Newtonian shear-thinning or pseudoplastic behavior as reported by Hesarinejad et al. [56] for *Plantago lanceolata* seed mucilage. This was also confirmed by the values of the power law index (n) presented in Figure 2B which, in all cases, was less than one.

Dias et al. tested the rheological and sensory properties of butter processed with different mixtures of cow/sheep milk cream and reported that sheep’s cream addition to cow’s cream improved the rheological properties of butters, originating lower consistency indexes and higher viscoelastic behavior, mainly due to the increased concentrations of short- and medium-chain fatty acids (SCFA; MCFA) and polyunsaturated fatty acids (PUFA). It was also reported that sheep milk cream butter was characterized by a greasy and striking aroma and a higher bitter taste [57]. These authors report values of 3981–7943 Pa for the elastic modulus (G′) and 1995–2511 Pa for the viscous modulus (G″) of pure sheep’s butter. The power law model parameters reported were (K = 71.8 ± 1.76 Pa.s^n−1^) and (n = 0.0814 ± 0.003), which are not comparable to those obtained by us, as they were obtained at 25 °C, while in the present work the rheological tests were carried out at 5 °C.

The results of the FA profiles, presented in Table 5, show some differences between butters, specifically with regard to the following FAs: C6-C14; C17; C18:1 c12; C18:2 c9t12; C20:3; C21:0 and C22:2. The fatty acid profile of the P butter presents significantly higher concentrations of short/medium-chain fatty acids (C4-C14). These differences, with impact on the total concentration of SFA, are most probably related to the metabolic activity of the microbial culture used. Underwood et al. report the production of short-chain fatty acids by probiotic bacteria, namely, *L. rhamnosus* and *L. acidophilus* [36]. Other authors also reported that the fermentation of saccharides, proteins, and peptides by intestinal bacteria originates SCFAs as end products, which appear to promote colonic homeostasis [58].

Significant differences between butters were observed regarding ∑MUFA and ∑PUFA concentrations, with the P butter having lower levels of MUFA and slightly higher levels of PUFA. Again, these differences are, most probably, the result of the metabolic activity of the mixture of lactobacilli used in the fermentation of cream.

As presented in Table 5, the sum of SFA was, in all cases, higher than 62%, with a significantly higher value obtained in the butter containing probiotic microorganisms (66.46%). Hilali and Rischkowsky [51] indicate that the predominant saturated and unsaturated FAs are palmitic acid (31.7–38.3%) and oleic acid (21.6–33.7%). Özkanli et al. [59] indicate the following ranges for pasteurized and non-pasteurized sheep’s ghee: C16:0 (29.83–31.30%); C18:0 (10.57–12.96%); C18:1 (31.08–32.67%). These results are in agreement with the ones obtained in the present work. 

Figure 3 presents the results of the microbiological analyses of butters and of the correspondent buttermilks. All butter samples produced with fermented cream presented levels of lactic acid bacteria higher than 6 Log CFU/g. The aromatic and the probiotics butters presented the highest levels of lactobacilli, which tended to increase until the 30th day, but decreased until the end of storage. Considering the counts of lactobacilli, it is clear that the P butter maintained adequate numbers of probiotic microorganisms until the end of storage (>7 Log CFU/g). Even in the case of frozen probiotic butter samples, the counts of lactobacilli were higher than 6 Log CFU/g over 90 days of storage (data not shown). Erkaya et al. [60] observed the survivability of selected probiotics during cold storage of butter produced with *Lactobacillus acidophilus* ATCC 4356 and *Bifidobacterium bifidum* ATCC 29521. They reported that butter with *B. bifidum* maintained its probiotic characteristics until the 30th day of storage. Olszewska et al. [61] also reported the cell viability of a *Bifidobacterium lactis* strain over four weeks in butter stored under refrigeration. Ewe and Loo [14] evaluated the properties of butter produced by cream fermented with *Lactobacillus helveticus* and reported significantly higher fat content and acidity. It was also noted that butter was softer than the conventional product due to increased levels of unsaturated fatty acids. These observations are contradictory with the ones obtained in the present work. 

Karaca et al. [62] report that kefir-cultured butter presented higher amounts of *Lactococcus* spp. by almost two log cycles and contained 5.24 Log CFU/g of *L. acidophilus*, while the control sample did not contain *L. acidophilus*. 

More recently, Bellinazo et al. [63] reported that the butter/bixin/isolated probiotic and butter/bixin/commercial probiotic formulations can be considered probiotic up to 74 and 69 days of storage, respectively, considering the concentration of ≥6 Log CFU/g.

Concerning yeast and mold counts, the aromatic and probiotic butters presented values below 4 Log CFU/g, while the sweet and the kefir butters presented values higher than 4 Log CFU/g, with a tendency to decrease over storage, in the case of the sweet butter. In kefir butter, whose starter contained the yeast *Debaryomyces hansenii*, the counts of yeasts and molds tended to increase until the 30th day of storage, decreasing afterwards. The same occurred with lactococci counts that reached values of >6.5 Log CFU/g at the 30th day of storage, but also decreased to ca. 5.0–5.5 Log CFU/g at the 60th and 90th days of storage. 

Concerning the microbial characteristics of buttermilks, higher values of lactobacilli and lactococci were observed for the samples containing the aromatic starter and probiotics (Figure 3D,E). In the latter, lactobacilli and lactococci counts were of the order of Log 10 CFU/mL at the beginning of storage, and levels higher than 8 Log CFU/mL were maintained during the 28 days of refrigerated storage. This observation indicates the excellent adaptation of those microorganisms to this matrix, which is an important factor regarding its use as a functional food.

The physicochemical characteristics of the buttermilks resulting from the different butters produced are presented in Table 6. Significant differences were observed regarding almost all components. Protein levels were of the order of 4–5% (*w*/*v*), and fat levels were between 2 and 3% (*w*/*v*). Surprisingly, the buttermilk resulting from the production of probiotic butter presented the lowest level of solids, while the total solids content was the lowest in the correspondent butter. To the lower amount of solids in butter it was expectable a higher amount of solids in buttermilk, but this was not the case. 

As it was expected, the pH of sweet buttermilk (>6) was higher than all others, which were around 4.5. The titratable acidity of the aromatic buttermilk and of the buttermilk containing probiotics was significantly higher than the ones of S and K samples. As it occurred in the case of butters, these observations confirm the good adaptation of the aromatic starter and of the mixture of probiotic bacteria both to butter and to buttermilk.

The evaluation of the viscosity of buttermilks (Table 6) showed significant differences between all products, with the highest value observed corresponding to the probiotics buttermilk. Again, a possible explanation for this fact can be the higher amount of saturated fatty acids in the P butter, and hence in the correspondent buttermilk.

The viscosity of sheep’s buttermilks produced with sweet cream was significantly lower than all others. Kefir and probiotics buttermilks presented the highest values. It can be concluded that, in both cases, the type of starter cultures played a significant role regarding this parameter. 

The color parameters of buttermilks also presented significant differences (Table 6). The product resulting from the aromatic butter presented clear differences, being darker (lower L* values) and with lower yellowness (lower b* values). 

Sakkas et al. compared the composition of sheep’s and cow’s sweet buttermilks and reported that the former was the most advantageous in terms of non-fat solids, protein and phosphorus contents. No significant differences were observed in the phospholipids content of ovine and bovine buttermilks. The antioxidant potential and emulsion stability of sweet ovine buttermilk were the highest [64]. These findings indicate the nutritional interest of sweet sheep’s buttermilk, which can be further enhanced by its fermentation with probiotic bacteria.

A recent paper by Sharma et al. evaluated the production of functional buttermilk and soymilk fermented by a *Pediococcus acidilactici* strain expressing the L-alanine dehydrogenase enzyme. The levels of total antioxidants, phenolics, flavonoids and especially L-alanine were significantly enhanced after LAB fermentation [65].

Ogrodowczyk et al. tested buttermilk-based formulations fermented with several lactic acid bacteria and *Bifidobacterium* strains. These authors report high growth rates for the strains tested. It was also indicated that the sensory quality of products was influenced by the profile of SCFA and free peptides. Two formulations, fermented with *L*. *bulgaricus*-151 and *Lactobacillus casei*-LcY, were the most advantageous, with desirable sensory, immunoreactive and biochemical properties [66]. 

Considering the continuous interest of researchers in the development of health-promoting products based on buttermilk, it can be inferred that this product can have potential in the European market, contributing to circular economy.

Regarding the sensory characteristics of butters (Appendix A) non-significant differences were observed between formulations, either by the preference test or by the ranking test. In general, all products were well accepted by the panelists. It is worth to note that the highest scores for aroma, taste and texture were obtained by the P and S butters at the beginning of storage. At the end of storage, all butter samples maintained good sensory characteristics. 

Pandya and Ghodke [1] found that traditional Turkish “yayik” butter made from goat’s milk had the most acceptable organoleptic characteristics when compared to “yayik” butters produced with sheep or cow milks. On the contrary, Hilali and Rischkowsky [55] report that sheep’s milk is preferred over goat’s milk for the manufacture of ghee because of the differences in flavor. Kahyaoğlu and Çakmakç [67] evaluated butter samples produced from cow’s, sheep’s and goat’s cream during a 90-day storage period at 4 °C, reporting that sensory analysis scores decreased in all the butter samples during the storage. The highest scores were obtained by the butter produced from cow’s milk.

As in the case of butters, non-significant differences were observed between buttermilks regarding sensory characteristics (Appendix A). In general, all products were well accepted by the panelists. It can be pointed out that both K and P buttermilks received higher scores for aroma, taste and consistency, indicating their good marketing potential. 

## 4. Conclusions

The results obtained agreed with previous works indicating the feasibility of producing butters and their correspondent buttermilks fermented with different starters and probiotic cultures. Moreover, the result regarding the viable cell counts of probiotic sheep’s butter and its correspondent buttermilk indicate their potential as carriers of probiotic microorganisms. Regarding the textural parameters of butters and the viscosity of the correspondent buttermilks, significant differences were observed between samples. These differences could be attributed to the differences in the fatty acid profiles of samples. It is important to highlight the differences observed in the FA profile of the probiotic butter, which can be interesting concerning health issues. However, further work is needed to evaluate this possibility. All products presented interesting scores regarding the sensory parameters evaluated, indicating their market potential. Additionally, the production of buttermilk containing probiotic bacteria can be an interesting strategy to valorize this by-product. Although European consumers are not familiar with buttermilk, its use as a vehicle for probiotic bacteria delivery can increase its attractiveness. In conclusion, it can be confirmed that the transformation of dairy products into competitive value-added commodities such as sheep’s butter and buttermilk can represent an opportunity to valorize peripheral rural areas.

## Figures and Tables

**Figure 1 foods-12-00331-f001:**
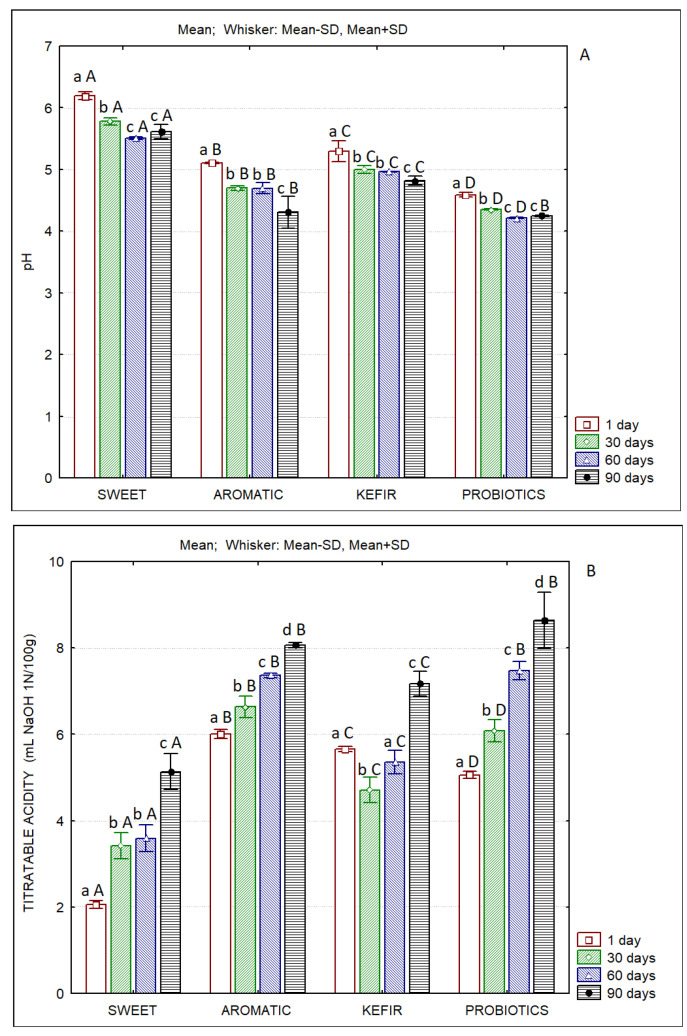
Evolution of pH (**A**) and titratable acidity (**B**) of sheep’s butters produced with sweet cream and with cream fermented with different starter cultures. Different lowercase letters (a,b,c,d) indicate significant differences (*p* < 0.05) between days of storage; Different capital letters (A,B,C,D) indicate significant differences (*p* < 0.05) between products at the same storage day.

**Figure 2 foods-12-00331-f002:**
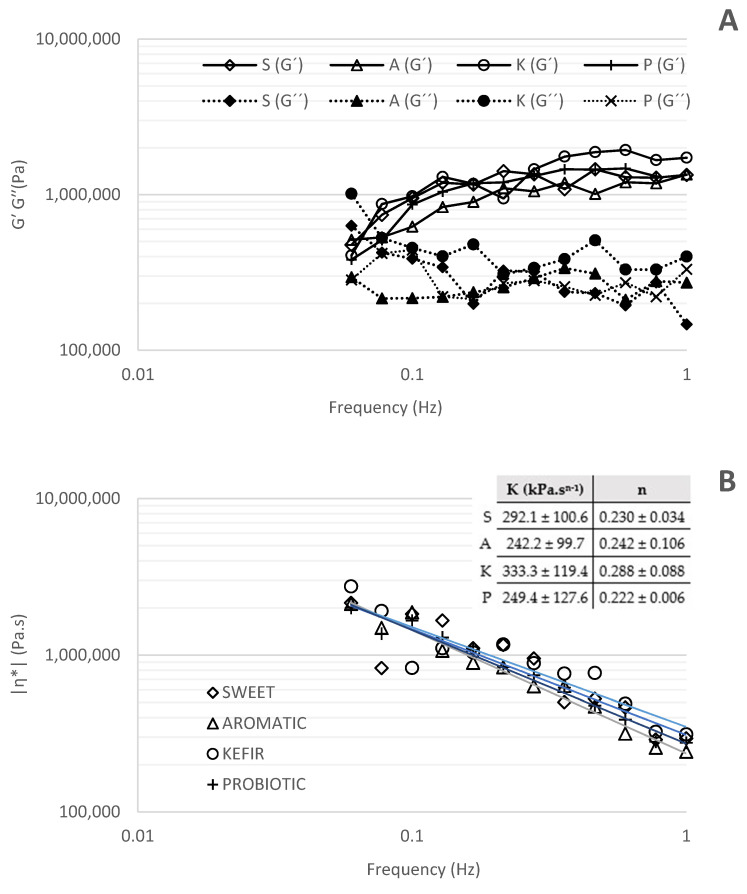
Rheological parameters of sheep′s butters produced with sweet cream (S) and cream fermented with aromatic starter culture (A), kefir (K) and probiotics (P). (**A**) Elastic (G′) and viscous (G″) moduli; (**B**) complex viscosity. According to power law model (Equation (2)) consistency index (*K*) and power law index (*n*).

**Figure 3 foods-12-00331-f003:**
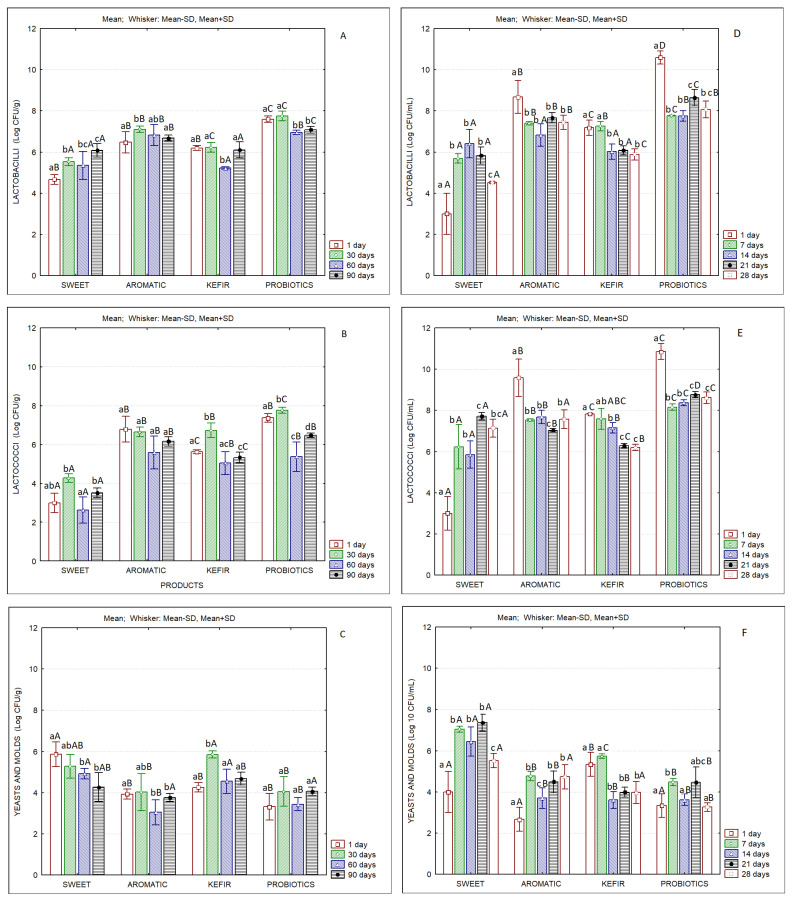
Microbiological characteristics of sheep’s butters and correspondent buttermilks produced with sweet cream and with cream fermented with different starter cultures. (**A**) lactobacilli in butter; (**B**) lactococci in butter; (**C**) yeasts and molds in butter; (**D**) lactobacilli in buttermilk; (**E**) lactococci in buttermilk; (**F**) yeasts and molds in buttermilk; different lowercase letters (a,b,c,d) indicate significant differences (*p* < 0.05) between days of storage; different capital letters (A,B,C,D) indicate significant differences (*p* < 0.05) between products at the same storage day.

**Table 1 foods-12-00331-t001:** Chemical composition of sheep’s butters produced with sweet cream and with cream fermented with different starter cultures. Different superscript lowercase letters in each column (a,b) indicate significant differences (*p* < 0.05) between butters.

Products	Dry Matter (% *w*/*w*)	Fat (% *w*/*w*)	Protein (% *w*/*w*)	Ashes (% *w*/*w*)
S—Sweet	87.24 ± 2.79 ^a^	83.63 ± 3.28 ^a^	0.34 ± 0.05 ^a^	1.19 ± 0.23 ^a^
A—Aromatic	87.57 ± 5.02 ^a^	83.42 ± 4.96 ^a^	1.08 ± 0.14 ^b^	1.35 ± 0.27 ^a^
K—Kefir	85.58 ± 4.92 ^a^	81.88 ± 3.89 ^a^	0.84 ± 0.19 ^b^	1.38 ± 0.23 ^a^
P—Probiotics	78.50 ± 3.13 ^b^	74.75 ± 2.18 ^b^	0.89 ± 0.11 ^b^	1.58 ± 0.26 ^a^

**Table 2 foods-12-00331-t002:** Color parameters (L*, a*, b*) of sheep’s butters produced with sweet cream and with cream fermented with different starter cultures, over storage (1 d, 30 d, 60 d and 90 d). Average values (±) standard deviation. Different superscript lowercase letters in each column (a,b,c,d) indicate significant differences (*p* < 0.05) between products regarding color parameters at a specific day.

	**L*1 d**	**L*30 d**	**L*60 d**	**L*90 d**
S—Sweet	90.00 ± 0.22 ^a^	88.00 ± 0.70 ^a^	81.23 ± 0.21 ^a^	84.67 ± 0.77 ^a^
A—Aromatic	90.93 ± 0.31 ^b^	85.63 ± 1.28 ^b^	81.73 ± 0.15 ^b^	83.75 ± 1.23 ^a^
K—Kefir	91.70 ± 0.17 ^c^	90.20 ± 0.42 ^c^	83.17 ± 0.71 ^c^	89.07 ± 0.45 ^b^
P—Probiotics	92.90 ± 0.17 ^d^	88.20 ± 0.72 ^a^	86.43 ± 0.64 ^d^	84.90 ± 1.44 ^a^
	**a*1 d**	**a*30 d**	**a*60 d**	**a*90 d**
S—Sweet	−3.00 ± 0.00 ^a^	−2.90 ± 0.08 ^a^	−3.03 ± 0.17 ^a^	−3.30 ± 0.29 ^a^
A—Aromatic	−3.30 ± 0.17 ^b^	−3.30 ± 0.28 ^b^	−3.33 ± 0.12 ^b^	−3.55 ± 0.31 ^b^
K—Kefir	−3.17 ± 0.25 ^ab^	−3.33 ± 0.10 ^b^	−3.43 ± 0.25 ^b^	−3.18 ± 0.43 ^a^
P—Probiotics	−3.20 ± 0.00 ^ab^	−3.27 ± 0.06 ^b^	−3.17 ± 0.35 ^ab^	−3.10 ± 0.35 ^a^
	**b*1 d**	**b*30 d**	**b*60 d**	**b*90 d**
S—Sweet	3.67 ± 0.17 ^a^	2.80 ± 0.22 ^a^	1.67 ± 0.31 ^a^	4.00 ± 0.13 ^a^
A—Aromatic	4.53 ± 0.21 ^b^	3.35 ± 0.97 ^b^	3.00 ± 0.10 ^b^	5.57 ± 0.27 ^b^
K—Kefir	4.63 ± 0.40 ^b^	4.70 ± 0.16 ^c^	4.50 ± 0.00 ^c^	6.87 ± 0.27 ^c^
P—Probiotics	4.80 ± 0.00 ^b^	4.30 ± 0.20 ^c^	4.00 ± 0.50 ^c^	6.44 ± 0.33 ^c^

**Table 3 foods-12-00331-t003:** Color differences ∆Eab* among products and over storage. (A) ∆Eab* between butter samples at each day of storage, 1, 30, 60 and 90. (B) ∆Eab* for the same butter between the first day of storage and days 30, 60 and 90 (vs. = versus).

**A—Storage Time**	**1 Day**	**30 Days**	**60 Days**	**90 Days**
	**Sweet**	**Sweet**	**Sweet**	**Sweet**
Aromatic vs...	0.99	2.77	1.12	3.03
Kefir vs...	2.00	4.64	6.15	14.58
Probiotics vs…	4.65	0.95	15.99	3.36
	**Aromatic**	**Aromatic**	**Aromatic**	**Aromatic**
Kefir vs...	0.29	9.13	3.28	16.57
Probiotics vs…	2.07	2.78	11.87	2.27
	**Kefir**	**Kefir**	**Kefir**	**Kefir**
Probiotics vs…	0.82	2.39	6.10	10.01
**B-Products**	**30 days**		**60 days**	**90 days**
Sweet 1st day vs...	2.58		38.88	14.80
Aromatic 1st day vs...	13.18		42.80	30.85
Kefir 1st day vs...	1.05		36.56	4.05
Probiotics 1st day vs...	11.35		21.25	32.95

**Table 4 foods-12-00331-t004:** Texture parameters of sheep’s butters produced with sweet cream and with cream fermented with different starter cultures. Average values and standard deviation (±). Different superscript lowercase letters in each column (a,b) indicate significant differences (*p* < 0.05) between products.

Products	Hardness (N)	Adhesiveness (N)	Gumminess (N)	Springiness	Cohesiveness	Resilience
S—Sweet	38.75 ± 7.30 ^a^	−7.16 ± 2.39 ^a^	10.80 ± 3.24 ^a^	0.85 ± 0.10 ^a^	0.29 ± 0.02 ^a^	0.09 ± 0.02 ^a^
A—Aromatic	30.78 ± 2.89 ^b^	−7.98 ± 1.81 ^a^	10.15 ± 1.77 ^a^	0.78 ± 0.12 ^a^	0.34 ± 0.04 ^a^	0.09 ± 0.01 ^a^
K—Kefir	37.52 ± 5.17 ^a^	−6.97 ± 1.00 ^a^	11.24 ± 2.15 ^a^	0.80 ± 0.08 ^a^	0.29 ± 0.02 ^a^	0.08 ± 0.02 ^a^
P—Probiotics	43.78 ± 12.43 ^a^	−5.46 ± 2.32 ^a^	12.66 ± 3.00 ^a^	0.73 ± 0.19 ^a^	0.27 ± 0.03 ^a^	0.10 ± 0.01 ^a^

**Table 5 foods-12-00331-t005:** Fatty acid profiles of sheep’s butters produced with sweet cream and with cream fermented with different starter cultures. Average values and standard deviation. Different superscript lowercase letters in each row (a,b,c) indicate significant differences (*p* < 0.05) between products.

Fatty Acids	S—Sweet	A—Aromatic	K—Kefir	P—Probiotics
C6:0	0.66 ± 0.02 ^a^	0.66 ± 0.00 ^a^	0.55 ± 0.00 ^b^	0.79 ± 0.00 ^c^
C8:0	0.88 ± 0.03 ^a^	0.87 ± 0.00 ^a^	0.77 ± 0.00 ^b^	1.07 ± 0.00 ^c^
C10:0	3.85 ± 0.11 ^a^	3.78 ± 0.01 ^a^	3.53 ± 0.00 ^b^	4.54 ± 0.00 ^c^
C12:0	3.28 ± 0.09 ^a^	3.22 ± 0.00 ^a^	3.13 ± 0.01 ^b^	3.69 ± 0.01 ^c^
C14:0	10.67 ± 0.29 ^a^	10.52 ± 0.01 ^a^	10.44 ± 0.05 ^a^	11.39 ± 0.03 ^b^
C14:1	0.19 ± 0.01 ^a^	0.18 ± 0.00 ^a^	0.18 ± 0.00 ^a^	0.19 ± 0.00 ^a^
C15:0	1.23 ± 0.02 ^a^	1.20 ± 0.02 ^a^	1.21 ± 0.01 ^a^	1.27 ± 0.00 ^a^
C15:1 c10	0.34 ± 0.00 ^a^	0.33 ± 0.00 ^a^	0.33 ± 0.00 ^a^	0.34 ± 0.02 ^a^
C16:0	32.24 ± 0.10 ^a^	31.60 ± 0.10 ^a^	31.80 ± 0.19 ^a^	32.53 ± 0.05 ^a^
C16:1 c7	0.26 ± 0.00 ^a^	0.26 ± 0.00 ^a^	0.26 ± 0.00 ^a^	0.26 ± 0.01 ^a^
C16:1 c9	1.22 ± 0.01 ^a^	1.20 ± 0.01 ^a^	1.21 ± 0.01 ^a^	1.27 ± 0.00 ^a^
C17:0	0.84 ± 0.03 ^a^	0.82 ± 0.00 ^a^	0.42 ± 0.57 ^b^	0.83 ± 0.00 ^a^
C17:1 c10	0.31 ± 0.01 ^a^	0.30 ± 0.00 ^a^	0.30 ± 0.00 ^a^	0.33 ± 0.02 ^a^
C18:0	10.56 ± 0.34 ^a^	10.25 ± 0.04 ^a^	10.39 ± 0.07 ^a^	10.18 ± 0.01 ^a^
C18:1 c9	25.47 ± 2.17 ^ab^	26.73 ± 0.09 ^a^	27.50 ± 0.18 ^a^	23.16 ± 0.12 ^b^
C18:1 c11	0.69 ± 0.01 ^a^	0.67 ± 0.00 ^a^	0.69 ± 0.00 ^a^	0.66 ± 0.00 ^a^
C18:1 c12	0.42 ± 0.02 ^a^	0.43 ± 0.00 ^a^	0.44 ± 0.00 ^a^	0.40 ± 0.00 ^b^
C18:2c9t12	0.42 ± 0.01 ^a^	0.41 ± 0.00 ^a^	0.41 ± 0.00 ^a^	0.44 ± 0.00 ^b^
C18:2	5.17 ± 0.02 ^a^	5.12 ± 0.02 ^a^	5.17 ± 0.04 ^a^	5.36 ± 0.02 ^b^
C18:3 c6c9c13	0.48 ± 0.01 ^a^	0.41 ± 0.12 ^a^	0.48 ± 0.01 ^a^	0.53 ± 0.00 ^a^
C18:3	0.06 ± 0.00 ^a^	0.06 ± 0.00 ^a^	0.07 ± 0.00 ^a^	0.06 ± 0.00 ^a^
C20:1	0.34 ± 0.01 ^a^	0.60 ± 0.39 ^a^	0.33 ± 0.00 ^a^	0.32 ± 0.00 ^a^
C20:3	0.15 ± 0.00 ^a^	0.14 ± 0.00 ^b^	0.14 ± 0.00 ^b^	0.13 ± 0.00 ^c^
C21:0	0.08 ± 0.00 ^a^	0.08 ± 0.00 ^a^	0.08 ± 0.00 ^a^	0.07 ± 0.00 ^b^
C22:0	0.05 ± 0.00 ^a^	0.05 ± 0.00 ^a^	0.05 ± 0.00 ^a^	0.05 ± 0.00 ^a^
C22:2	0.07 ± 0.00 ^a^	0.06 ± 0.00 ^b^	0.06 ± 0.00 ^b^	0.06 ± 0.00 ^b^
C24:0	0.05 ± 0.02 ^a^	0.05 ± 0.00 ^a^	0.05 ± 0.00 ^a^	0.05 ± 0.02 ^a^
C24:1	0.03 ± 0.00 ^a^	0.01 ± 0.02 ^a^	0.04 ± 0.00 ^a^	0.02 ± 0.00 ^a^
∑SFA	64.38 ± 1.92 ^a^	63.09 ± 0.14 ^a^	62.38 ± 0.024 ^a^	66.46 ± 0.12 ^b^
∑MUFA	29.26 ± 0.27 ^a^	30.71 ± 0.27 ^a^	31.28 ± 0.20 ^a^	26.96 ± 0.11 ^b^
∑PUFA	6.36 ± 0.20 ^a^	6.19 ± 0.14 ^a^	6.34 ± 0.05 ^a^	6.58 ± 0.01 ^b^

SFA = saturated fatty acids; MUFA = monounsaturated fatty acids; PUFA = polyunsaturated fatty acids.

**Table 6 foods-12-00331-t006:** Physicochemical characteristics of sheep’s buttermilks produced with sweet cream and with cream fermented with different starter cultures. Different superscript lowercase letters in each row (a,b,c,d) indicate significant differences (*p* < 0.05) between products. (T.A. = titratable acidity).

Parameters	Products			
	S—Sweet	A—Aromatic	K—Kefir	P—Probiotics
Total solids (% *w*/*w*)	13.92 ± 0.13 ^a^	11.87 ± 0.11 ^b^	13.70 ± 0.11 ^c^	10.49 ± 0.02 ^d^
Protein (% *w*/*w*)	5.12 ± 0.00 ^a^	4.25 ± 0.10 ^b^	3.70 ± 0.07 ^c^	3.95 ± 0.05 ^d^
Fat (% *w*/*v*)	2.57 ± 0.06 ^a^	2.53 ± 0.12 ^a^	2.93 ± 0.58 ^a^	2.10 ± 0.00 ^b^
Ashes (% *w*/*w*)	1.52 ± 0.11 ^a^	0.68 ± 0.07 ^b^	0.78 ± 0.33 ^b^	0.76 ± 0.10 ^b^
T.A. (mL NaOH 1 N/L)	35.10 ± 0.47 ^a^	84.98 ± 0.64 ^b^	56.3 ± 0.49 ^c^	84.75 ± 0.90 ^b^
pH	6.24 ± 0.01 ^a^	4.65 ± 0.00 ^b^	4.92 ± 0.01 ^c^	4.52 ± 0.01 ^d^
Viscosity (cPs)	18.42 ± 1.42 ^a^	43.08 ± 4.31 ^b^	165.77 ± 1.97 ^c^	219.87 ± 8.81 ^d^
Color values				
L*	82.07 ± 0.15 ^a^	77.00 ± 0.35 ^b^	83.70 ± 0.10 ^c^	82.40 ± 0.26 ^a^
a*	−3.97 ± 0.15 ^a^	−3.57 ± 0.25 ^b^	−3.83 ± 0.06 ^b^	−4.20 ± 0.10 ^c^
b*	4.27 ± 0.21 ^a^	0.97 ± 0.06 ^b^	5.97 ± 0.06 ^c^	5.33 ± 0.21 ^d^

## Data Availability

Most data are available in the manuscript. Specific information will be provided upon request to the corresponding author.

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
