# Peer review of "Sheep’s Butter and Correspondent Buttermilk Produced with Sweet Cream and Cream Fermented by Aromatic Starter, Kefir and Probiotic Culture"

_foods, 2023, doi:10.3390/foods12020331_

Round 1
Reviewer 1 Report
Enhancing the nutritional/economical value of small/medium size dairy enterprises products is important for their survival in a competitive market. Probiotic fermented dairy products is a trend driven by consumer´s acceptance of such products.
Probiotic dairy products are products that contain living probiotic micro-organisms in an adequate matrix and in sufficient concentration, so that after their ingestion, the postulated effect is obtained and is beyond that of usual nutrient suppliers.
Sheep production contribute to the livelihoods of a large number of small-scale resource-poor farmers in Middle Eastern countries. The main dairy products from sheep are yogurt and cheese, mostly produced by households and small-scale local processors using traditional methods.
The purpose of this study was to develop different sheep´s butter samples and the correspondent buttermilks after fermentation of the cream by mesophilic aromatic starter, kefir culture and a mixture of probiotic bacteria.
Nevertheless, quality and safety aspects related to small scale processing have not been thoroughly studied and require more attention in order to sustain or improve the market access of small processors.
Recommendations
L 113 Please mention the source of the raw material milk.
L 115 Please specify the type of location used as experimentation venue (on-site, laboratory, pilot plant etc.)
L 156 Please present the method used for protein analysis.
L 250 Please present the data in fig. 1 in a table form, for better comprehension.
Author Response
Dear reviewer. Thank you very much for the carefukk revision of the manuscript and for all the sugestions that helped us improve its quality. Please find attached our responses to all the questions indicated by reviewers.

Reviewer 2 Report
Please find attached

Author Response

(The authors gave the same response as above.)

Reviewer 3 Report
The work demonstrates the application of various starter cultures on pre-fermentation of cream before subjecting to butter manufacture. The physicochemical, biochemical and sensory characteristics of butter and butter milk as by-product were investigated. The manuscript was written with appropriate English language level. In my humble opinion, however, several major points as indicated below should be clarified and revised before the decision can be made.
· Line 33-102: I would recommend shortening the introduction section. Much information regarding the nutritional aspects which were not included in the present study should be omitted. Information regarding potential technological development of butter and butter milk as a probiotic carrier along with by-product valorization should be better highlighted.
· Line 118-130: Please clarify that the inoculation rate at 5% resulted in which level of initial bacterial numbers (CFU/g) at the starting point of fermentation?
· Line 242: What would be the reason for this significantly lower yield influenced by the aromatic starter fermentation?
· Line 253-261: What would be the reason for post-acidification found in the sweet, un-fermented, samples?
· Line 278-280: Please revise the usage of statistically significant labels in Fig. 3. It is unclear.
· Line 267-276: It should be clearer if the authors can demonstrate the color of products using pictures incorporated with Table 1 and 2.
· Line 292-297: The authors should discuss what would be the reasons for different rheological patterns among samples derived from different types of fermentation in this study.
· Line 306-315 and Table 3: This part is a very interesting result. The authors should carefully look through these data and highlight what were the differences in FA profiles of products derived from different starter cultures and discussion linked to their metabolisms and lipolysis activities. The discussion could be also linked to the lipolysis-related flavor compound formation in the products.
· Figure 7: I suggest to revise the Fig. 7. in a Table form with ranking scores ± SD. This would be easier to understand for the readers.
· Line 370-417: The results of butter milk should be better merged with the same type of analysis for butter. Please kindly re-structure your manuscript.
· Overall discussion: I would recommend that the authors should better discuss the influences of different starter cultures, i.e. lactic acid bacteria and/or with yeast consortia, applied on the overall biochemical characteristics and sensory properties of products found in this study.
· Conclusion: The authors mentioned about the butter by-product valorization, however, there was no issue on this point addressed and discussed in the results and discussion section.
Author Response

(The authors gave the same response as above.)

Reviewer 4 Report
The authors presented the physicochemical, microbiological, and sensory properties of sheep´s butter and correspondent buttermilk produced with sweet cream and cream fermented by the aromatic starter, kefir, and probiotic culture. The idea of the article is good. But unfortunately, it is not well designed and the analyzes performed have not been scientifically and accurately analyzed. The results have not been well discussed. However, a few points to improve the current format of the article will be mentioned below:
The abstract should be more informative by giving real results rather than elastic sentences. Important and main contents should be given. Support the results with some quantitative data. Moreover, no conclusions are provided.
Materials and methods: “2.3.Physicochemical analyzes” should be separated, and each test should be given a separate title under it.
Be careful in numbering the titles.
There are too many figures and tables. Figures 1, 2, 3, and 4 should be merged and displayed as a table.
Analyzes related to rheological properties are very incomplete and need proper analysis.
The graphs shown in Figure 5 should be logarithmic and the frequency dependence should be checked with the power law model.
Figures 7 and 9 must be displayed in another way.
The results and discussion section should be thoroughly revised and require an in-depth analysis of the results.
Conclusion: what is the future of your findings? The conclusion is not insightful, what are your suggestions?
Author Response

(The authors gave the same response as above.)

Round 2
Reviewer 2 Report
This revision has significantly improved the manuscript. No further comments from the reviewer.
Author Response
Thank you for your carefull revision of the manuscript.
Reviewer 3 Report
The revised version of manuscript is now satisfied and can be acceptable for publication.
Author Response

(The authors gave the same response as above.)

Reviewer 4 Report
The corrections made are good. But still in the rheological properties section, apparently you did not understand my request and besides not logarithmizing the viscoelastic plot, the dependence of the viscoelastic moduli on frequency was not specified. It is necessary to have a proper discussion on the data obtained from fitting the variable frequency sweep data with the power law model. Please see these articles to make my explanation clearer (Figure 2).10.1016/j.foodhyd.2013.07.017 ; 10.1016/j.ijbiomac.2017.10.102 ; 10.1016/j.ijbiomac.2019.10.093 ; 10.1186/s40538-022-00322-2
The number of figures and tables is still too much. However, many figures can be merged. The data mentioned in some tables can also be given only in the text of the article.
It may be better to include some figures in the supplementary data.
The significant letters given in the tables should be specified whether it is for the data of a row or a column!? (Table 7)
The significant letters of the data should be specified (Table 6). Decimals (.) should be written correctly (0.69 not 0,69).
Are you sure of the data units listed in Table 4? 43 N for the hardness of butter tissue, is it correct?
Author Response
The corrections made are good. But still in the rheological properties section, apparently you did not understand my request and besides not logarithmizing the viscoelastic plot, the dependence of the viscoelastic moduli on frequency was not specified. It is necessary to have a proper discussion on the data obtained from fitting the variable frequency sweep data with the power law model. Please see these articles to make my explanation clearer (Figure 2).
Many thanks for your help and suggested papers. As you recommended, graphs of G´, G´´ and complex viscosity in terms of frequency have been plotted on a logarithmic scale. The discussion on the viscoelastic behavior of butters was improved according to the Power-Law model (G´=agb). The damping factor (tan d) was also evaluated and discussed, which complemented the analysis, however the graph (tan d) versus frequency was not shown because it can be inferred from the graph G´and G´´.
The number of figures and tables is still too much. However, many figures can be merged. The data mentioned in some tables can also be given only in the text of the article.
It may be better to include some figures in the supplementary data.
Figures showing the evolution of microbial counts on butter and buttermilk samples were fused (previous figures 3 and 4 are now figure 3).
Table 7 with data of sensory analysis of butter and buttermilk was placed as supplementary material. No letters were placed in these table since in both butter and buttermilk samples, non-significant differences were observed regarding sensory scores.
The significant letters given in the tables should be specified whether it is for the data of a row or a column!? (Table 7)
Corrected as proposed in all tables in which superscript letters were used to identify differences.
The significant letters of the data should be specified (Table 6). Decimals (.) should be written correctly (0.69 not 0,69).
Corrected as proposed in all tables in which superscript letters were used to identify differences.
Are you sure of the data units listed in Table 4? 43 N for the hardness of butter tissue, is it correct?
Yes, this is the correct value. In the first manuscript submitted the figure presenting the texture analysis had an error. In the text was stated that the P butter had the highest hardness, and this was not reflected in the correspondent figure. When substituting the figure by the table the error was detected and corrected. The temperature at which the tests were performed (7ºC) all samples presented high hardness values. Perhaps, using a temperature of 10-12 ºC for the tests, would better reflect the conditions of consumption of butter. On future works this aspect will be considered.